# Chiroscript: Transcription System for Studying Hand Gestures in Early Modern Painting

**Temenuzhka Dimova**

Laboratory for Cognitive Research in Art History (CReA Lab), Institute of Art History, Vienna Cognitive Science Hub, University of Vienna, 1090 Vienna, Austria; temenuzhka.dimova@univie.ac.at

**Abstract:** The main goal of this article is to introduce a new method for the analysis of depicted gestures in painting, namely a transcription system called chiroscript. Based on the model of transcription and annotation systems used in linguistics of co-speech gestures and sign languages, it is intended to provide a more systematic and objective study of pictorial gestures, revealing their modes of combination inside chirographic accords. The place of chirograms (depicted hand gestures) within pictorial semiotics will be briefly discussed in order to better explain why a transcription system is very much needed and how it could expand art historical perspectives. Pictorial gestures form an understudied language-like system which has the potential to increase the intelligibility of paintings. We argue that even though transcription is not a common practice in art history, it may contribute and even transform semiotic analyses of figurative paintings.

**Keywords:** chirogram; hand; gesture; transcription; annotation; painting

## 1. Introduction

Depicted hand gestures play an essential but not yet theoretically clarified role within the field of semiotics of art. To correctly build the foundations for studying pictorial gestures, it is necessary to firstly recognize that in addition to being components of painting, they are related to another scientific field: the study of meaningful human gestures. The study of meaningful human gestures itself is part of the semiotics of language, namely in co-speech gestures studies or in the linguistics of sign languages. Clearly, there are two paths by which we could apply semiotic perspectives to the study of pictorial hand gestures: either through the traditional semiotics of art, where depicted hands are a "shape" or a "form" (Carter 1976; Bergesen 2000), equal to any other element of the painting, or through the semiotics of language, in which they would be a distinctive manifestation of human gestures, remarkably emerging within the ecosystem of aesthetic compositional rules.

In this article, we argue in favor of the second possibility. The understanding of pictorial gestures will significantly improve if they are studied through the perspective of the semiotics of language. But the available methods in art history do not allow a sufficiently rigorous and objective analysis of the pictorial gesture syntax. Therefore, we propose the introduction of a new method: a transcription system called chiroscript. Before describing this new method, we will present the pictorial chirograms with their inscription within specific semantic categories and their capacity to form chirographic accords. We will also briefly discuss the close relationship between chirographic accords and pictorial semiotics. These two introductory passages are necessary to better understand why a transcription system is needed in art history. Then we will describe the chiroscript and give some examples of transcription, specifically for representations of the Last Supper. More specific attention will be drawn to three versions of this subject by Domenico Ghirlandaio. Finally, we will discuss the further development of the chiroscript and its interdisciplinary implications.





## 2. Pictorial Chirograms

Early Modern paintings contain a repertoire of around 120 hand gestures called chirograms (Knowlson 1965; Barasch 1987; Muchembled 1987; Bremmer and Roodenburg 1991; Kirchner 1991; Jacobus 1994; Dalli Regoli 2000; Chastel 2001; Rehm 2002; La Porta 2006; Dimova 2019; Niccoli 2021). Their interpretation is crucial for the understanding of the depicted scenes and, on a greater scale, for the analysis of iconological traditions. According to the pictorial genre and the specific content they transmit, the chirograms belong to the following semantic categories: discursive, emotional, devotional, ritual, insulting, postural, and manipulating (see table in Appendix A).

When the characters are involved in an active speaking attitude, they perform different forms of discursive gestures. Some of them express a teaching or an argumentation such as the comput digitis, typical for the representations of philosophers, orators, and Christ. All the deictic gestures (pointed forefinger, pointed thumb and pointed open hand) are also part of this category, assuring the coordination of personal, spatial, and temporal references inside the pictorial narratives. During the act of speech, the depicted characters may interact with each other or address the beholder directly. A typical gesture of inner interaction is that which consists of putting a hand on someone else's shoulder, while a gesture of mediation with the viewer may be, for example, the signum harpocraticum, inviting silence and revealing the secret nature of some element in the painting. Rare but important to mention are also all the chirograms inspired by ancient numerical dactylology (systems of representation of numbers with the fingers). Some characters, such as Saint John the Baptist, may display numbers to evoke a theological or a philosophical concept or to literally speak about some quantitative unity.

Another semantic category of chirograms includes all those related to the emotions and inner states of the characters. In scenes like the Crucifixion or the Lamentation, it is common to see intertwined fingers with the hands twisting in ascendant or descendant positions in order to express deep sorrow and despair. Other emotional chirograms are the tight fist, denoting physical pain or anger, and the palms raised forwards because of astonishment. In this same semantic category can also be found all the gestures related to some strong stimulation of the senses, provoking an attitude of protection. Typical examples are the hands covering the nose in the Raising of Lazarus or the gesture of glare protection in the scenes of Conversion, Ascendance, or Transfiguration.

The third semantic category regroups the devotional chirograms of Christian iconography. They are neither discursive nor emotional but rather encapsulate some deep state of prayer, veneration, or spiritual meditation. This group of gestures is largely spread because of the frequency of religious topics in Early Modern iconography. Open hands raised up, hands crossed on the chest, joined palms, and the act of lifting the veil or exhibiting the stigma are some well-known examples of devotional chirograms.

A similar but not necessarily religious semantic category is that dedicated to ritual gestures. Mainly related to political and social contexts, they intervene to seal oaths, pacts, and vows of allegiance or to denote some other sociocultural practice. Breaking a staff on one's knee, the dextrarum junctio, the iunctor gesture, and putting on a wedding ring can be typically seen in representations of marriages, for example. Ritual chirograms are a good testimony of social traditions beyond the context of art.

The fifth semantic category contains insulting and diverting chirograms, mainly appearing in genre paintings and in some religious topics such as the Mocking of Christ. They are rare but are quite various and instructive about local gesture traditions. Interesting examples are the thumbs-up, the *manichetto*, the fica, and the forefingers crossed in the air. Some of these chirograms may surprisingly appear in sacred iconography. The apotropaic use of blowing the nose in the Last Supper is an example. Alternatively, they may appear only once, as the jester gesture in David Ehrenstrahal's "Portrait of a court jester" (1652, Nationalmuseum of Stockholm).

The postural chirograms correspond to more passive attitudes. They bring information about the identity and the status of the characters, about specific virtues and vices

or civic manners. They may also be created by painters for stylistic reasons. Finally, the gestures which consist in manipulating some objects also have to be considered since they may be actively involved in the symbolic and narrative purpose of the painting.

The context in which the chirograms appear is crucial for their precise definition. Some gestures may have similar shapes but different meanings. The dactylological chirograms, for example, look like other much more frequent gestures and may be easily misinterpreted. Just like in spoken languages there are homonyms and in sign languages there are homosigns, in painting, there are homochirograms in which the configurations of the hand are identical but the meanings are completely different. A striking example is the offensive horns gesture (mano cornuta) which might also appear as a version of the benedictio in 14th century representations of Saint John the Baptist. In contrast, one and the same chirogram can appear with different shapes because of stylistic variations, to the point that it can become almost unrecognizable. The comput digitis, for example, possesses high stylistic diversity.

In addition to the homochirograms and the stylistic variations, a frequent phenomenon in painting is ambivalent chirograms. These are hand shapes which refer to two different gestures by figuring something in between. For example, the benedictio could in some cases look also like an open hand or a pointed forefinger. The ambivalent chirograms are not just stylistic caprices. They contain a semantical depth which is to be taken into account during the analysis.

The identification of each chirogram in a particular painting is an important step but it is not sufficient. To be systematic, the analysis of chirograms needs to be anchored in the general principle of pictorial composition, in which every element counts and contributes to the whole together (Dimova 2019). Early Modern written sources related to the pictorial techniques insist on the notions of harmony and union of all parts as conditions for quality (Fréart de Chambray 1662; de Piles 1668; Félibien 1676; Restout 1681). The term "accord" is used for the successful combination of both colors and figures. From this perspective, we can say that in each painting, gestures of different types together form a chirographic accord which conveys a general meaning going beyond the sense of every single chirogram. If the hands have been conceived as a "whole" by the painter, they need to be examined as such in the same way that linguistic analysis takes into account every single part of a sentence. If we selectively direct our attention only to some chirograms in one pictorial composition and neglect others, it betrays the notion of the accord. Investigating combinations, associations, and interactions of chirograms can reveal the syntax of this frozen but prolific gesture language.

Thus, if we take the general definition of each chirogram and put it in the context of the chirographic accord, a much more precise meaning will be revealed. In every pictorial scene, the characters form all together a specific discourse by their gestures, which can even be translated into words if rigorously analyzed. Chirographic accords of different paintings can be also compared to each other in order to reveal the underling syntax of gestures in Early Modern iconography. But all these steps are only possible if we acknowledge the fundamental belonging of chirograms to the semiotics of language.

## 3. Chirographic Accords and Pictorial Semiotics

The comparison between language and painting is frequent in semiotic discourses, in which authors systematically identify equivalents or analogies. For example, the phonemes of languages are dots or lines in paintings, the sentences are the whole composition, etc. (Carter 1976; Bal and Bryson 1991; Bergesen 2000; Sarapik 2013). As relevant as it can be, this framework does not take into account the fact that inside figurative paintings, there is something that speaks. Unlike the occasionally inserted inscriptions, hand gestures, facial expressions, and body postures constitute an organically implemented metalinguistic element in the paintings. In fact, the comparison between art and language is often based on the opposition between a visual and an auditory system, suggesting that human language is only transmitted by sounds (and their direct graphic representations in writing). This

point of view, however, implies ignoring the whole visual family of sign languages. The problem is completely external to art history. It reflects the lack of scientific recognition of sign languages until the 1970s. Since then, an increasing number of linguistic studies have been dedicated to the grammar and syntax of these visuo-spatial dynamic languages, today accepted by the academic communities as equal to the auditory–phonetic languages (Stokoe 1972; Emmorey 2023). Consequently, art historians may want to update the semiotic framework of figurative art and include comparisons between static pictures and the imagery of visual languages. It is not impossible that they share, to some degree, particular iconic patterns.

It is argued in this article that every chirogram works as a basic unit and a signifier, directing the viewer towards some specific meaning, while the combination of all these meanings gives the general semantic orientation of the chirographic accord. If we apply Charles Morris's semantic diagram to pictorial gestures, the hand configuration of, for example, benedictio is the sign vehicle, while the teaching of the Logos is the denotatum. The relationships between all the parts of the chirographic accords reveal the syntax of this language, and the beholder of the painting is the interpretant. In general, none of these four constituents is permanent: the sign vehicles have multiple stylistic variations, the denotata must always be slightly adjusted according to the context, the syntax obviously differs in every composition, and the interpretant may belong to different categories, namely historical beholders, empirical beholders, ideal beholders, experts, non-experts, etc. (Bal and Bryson 1991). These multiple inputs give the impression that works of art do not form a single and unequivocal language. As reported by Sarapik (2013), the skeptical view on the relationship between art and language is based on the lack of double articulation. However, the position defended here is that pictorial gestures do form a language-like system which is not identical to dynamic nonverbal communication but do possess a semantic structure and double articulation.

Furthermore, pictorial chirographic accords are characterized by an important combinatorial flexibility. According to Bergesen (2000), Chomsky's theory about generative grammar could be applied to art. "Language and art may both operate as discrete combinatorial systems governed by a set of principles which allow permissible combinations of linguistic and artistic primitives", he said. The chirograms can be combined in infinite ways and, despite the iconographic prototypes painters follow, every accord is unique. Even paintings with binary accords, such as those of the Annunciation, contain important gestural diversity (Niccoli 2021). In iconographic topics containing more than two characters, the accords tend to be unique inventions. Painters systematically avoid reproducing all the attitudes of a previous model; in other words, every painting is a specific sentence produced from the painter's mind in the same way as unprecedented linguistic sentences are generated constantly in human speech.

## 4. Chiroscript: Mimetic Symbols and Linearity

One way of assuring a precise semantic study of the chirographic accords is to provide a transcription system for the chirograms. Using such a system would force the researcher to analyze every depicted hand consciously as being part of the general composition and to consider it in a more objective way. Usually, it is not easy for the analytical gaze to be objective when looking at a painting. There are many distractive factors, such as the compositional complexity and the stylistic atmosphere of every single artwork. Semantic similarities and gestural patterns between paintings can be missed if we only observe the composition as its organic whole. Gestures must be extracted and analyzed independently. This is the main scope of the chiroscript: providing a separate visualization of the chirographic accord with the aim of analyzing its structure and semantic properties.

As stated above, the iconicity of the transcription is an important factor. Therefore, we decided to look for mimetic symbols whose shape reminded us somehow of the shape of the chirograms. We found such visual correspondences in the Unicode Standard of the following alphabets: Armenian, Latin, Cyrillic, Greek, and Coptic. For example, the Ar-

menian capital vew—Ⴁ—can evoke an open extended hand. The hands crossed on the chest can be suggested by the Latin small gamma—ɣ—while the pointed thumb could be figured by the Greek small sigma—σ. Some of the selected symbols may look less mimetic than others, but in general, we tried to follow this rule as much as possible. Two of them do not belong to any alphabet: the section sign—§—for figuring the intertwined fingers and the infinity sign—∞—for the dextrarum juncto. We attributed a specific symbol only to the 22 most frequent and/or significant chirograms. They are all presented with illustrations at the end of this article (see Appendix B). Being widespread and significant, these 22 symbols are regularly used for the transcription. But outside of them, there are many other gestures which appear more occasionaly in painting, as discussed above. For their annotation, we propose the symbol Ş, (meaning specific) followed by an alphabetic abbreviation in superscript. A scalable list of these specific gestures has been established (see Appendix C). A similar approach was adopted for the manipulation of objects—they appear in the chiroscript as the symbol ꞥ followed by an alphabetic abbreviation (see Appendix C). These two lists—the specific chirograms and the chirograms of manipulation—are extensible and can be adapted to various case studies. The transcriber can define new abbreviations for every new gesture.

A distinctive symbol—Ɔ—has been assigned to gestures which are hard to identify due to problems of visibility, some ambiguity of the configuration, or simply because they are unknown. There is also a symbol for a hand in rest—♭—helping to distinguish between active and inactive gestures. One very frequent type of chirograms—the deictics—need supplementary precision in the chiroscript, namely the direction of pointing. This will be specified by the four directional arrows in superscript, (Unicodes 2190, 2191, 2192, 2193).

To simplify the transcription, we created the Chirokey, a customized keyboard which can be downloaded and immediately applied (see link to Supplementary Material). The chirographic symbols are typed with this keyboard while the Latin alphabetic abbreviations in superscript are typed with a normal keyboard. Therefore, the transcription requires constant switching between two keyboard languages. A few practicing makes this procedure quite easy and quick. Most of the 30 letters of the Chirokey are located on the Latin keys of the physical keyboard (QWERTZ) which have some visual similarity with them, in order to facilitate the learning process of the typing (see table of key correspondences in Supplementary Material). The font Times New Roman is recommended to best preserve the morphological resemblance with the chirograms. For more detailed technical guidance for the transcription procedure, a user guide can be consulted (see link to Supplementary Material).

Here is a schema (Figure 1) presenting the transcription of Raphael's "Transfiguration" (Figure 2) and summarizing the main characteristics of the chiroscript which will be discussed in the following sections. The linearity of the system is one of its main features. This means that the depth planes of the pictorial composition are not taken into account. The characters are annotated in a wave-like movement, and they ultimately appear in the transcription in one line. If there are different registers in the painting, as here, they appear in the chiroscript on different lines.

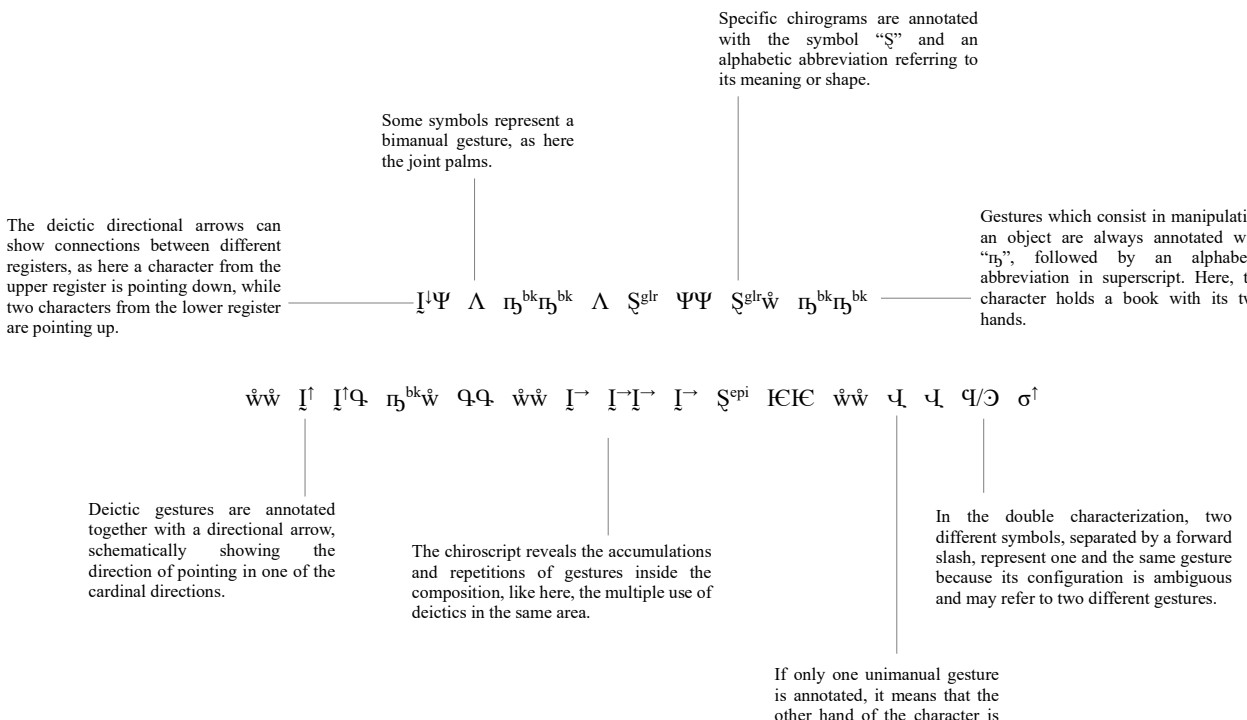

**Figure 1.** Transcription of Raphael's "Transfiguration" and main features of the chiroscript.

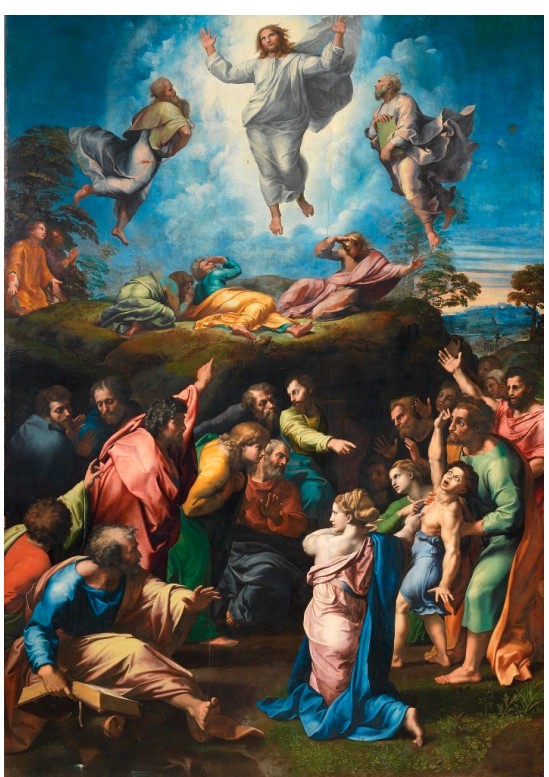

**Figure 2.** Raphael, "Transfiguration", 1518–1520, oil on panel, 410 × 279 cm, Pinacoteca Vaticana, Vatican City.

Transcription practices have already been implemented in the neighboring disciplines, namely the study of co-speech gestures and sign languages. The creation of annotation and transcription tools for gestures has been a focus of research interest since the 1950s, and it became a growing area over the last two decades (Birdwhistell 1952; Stokoe 1960; Slobin et al. 2001; Boutet and Garcia 2006; Kipp et al. 2007; Kato 2008; de Courville et al. 2010; Garcia and Sallandre 2013; Pak-Hin Kong et al. 2015; Bianchini et al. 2018; Boutet et al. 2018; Power et al. 2022). The need to transcribe gestures and signs in order to improve or even make possible their study has been regularly underlined by researchers. According to Slobin, "the goal of all transcription is to produce a permanent, written record of communicative events, allowing for analysis and re-analysis. […] The most basic aim of every system of notation of behavior is to help researchers see patterns in the data" (Slobin et al. 2001).

The creation of an effective annotation system for visual dynamic languages and communicative systems is extremely challenging (Garcia and Sallandre 2013). The overlapping of multiples parameters in these languages tend to resist graphic representations. In this sense, art historians may consider themselves lucky given that they deal with static pictorial characters. The depicted body attitudes still possess several parameters, related to their location, orientation and degree of deployment, but their two-dimensional characteristics significantly facilitate the efforts of transcription. Moreover, what really matters in the analysis of pictorial gestures is their objectivation. In order to be able to see the patterns in the chirographic accords, we need to highlight the main typological categories of the chirograms and step away from their almost infinite stylistic variety. Also, they will be presented in a linear fashion without consideration of the depth planes. If included in the transcription, the spatial and stylistic complexity of the chirograms could overcharge the system and make it ineffective.

## 5. Visualizing the Properties of the Chirographic Accords

The chiroscript is intended to enhance the visibility of the following semiotic aspects of the accords:

- Predominant semantic categories
- Co-occurrent chirograms
- Ambivalence and clarity
- Repetitions and spatial symmetries
- Deictic structures
- Chronotopic divergences
- Ratio between active and inactive hands
- Rare, unexpected, and unknown chirograms

To illustrate these properties, we will comment on the transcriptions of several representations of the Last Supper, namely by Philippe de Champaigne, Leonardo da Vinci, and Domenico Ghirlandaio. Here is the example by Champaigne (Figure 3):

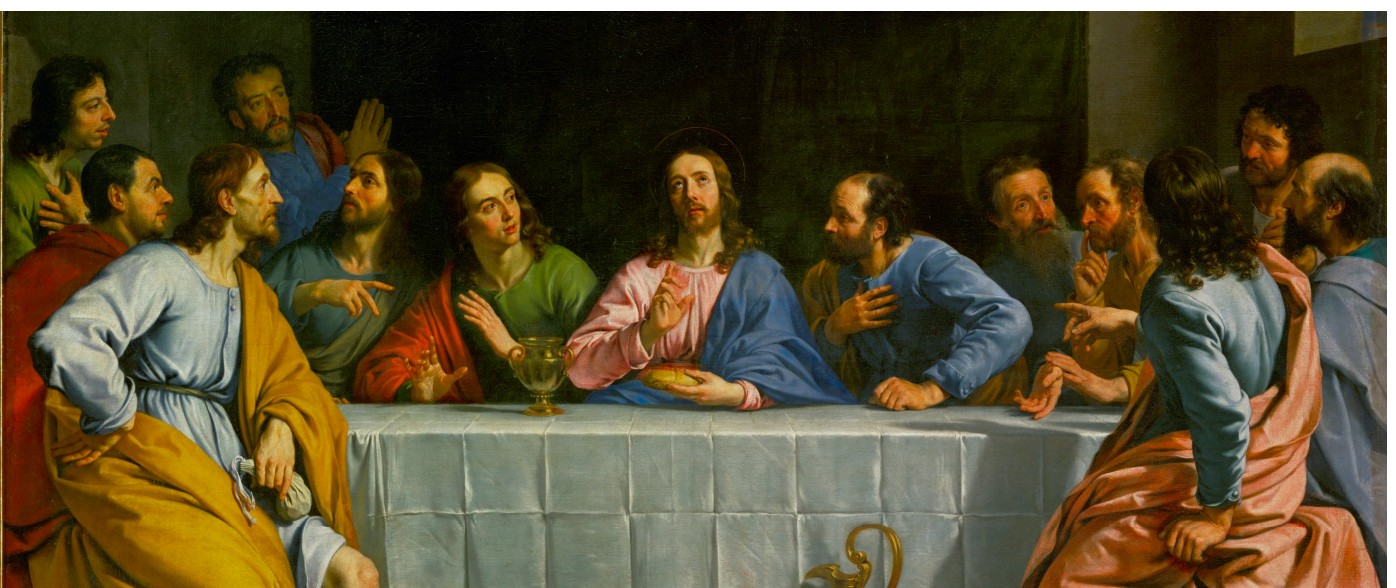

**Figure 3.** Philippe de Champaigne, "The Last Supper", (detail), c. 1662, oil on canvas, 158 × 233 cm, Louvre, Paris.

| ɣ | Ṣ<sup>hip</sup>ꜧ<sup>mo</sup> | Λ | Ị<sup>→</sup>ꜧ<sup>f</sup> | ẘẘ | ʮ/Ị<sup>↑</sup>ꜧ<sup>br</sup> | ૧○ | ५ | ʔʝ | Ị<sup>←</sup>ꜧ<sup>f</sup> | § |
|---|---|---|---|---|---|---|---|---|---|---|
| 1 | 2 | 3 | 4 | 5 | 6 | 7 | 8 | 9 | 10 | 11 |

There are 11 characters with visible hands and 18 chirograms, composing the general accord of the story. While commenting on the transcription, it is possible to refer to the position of each character ("ch.") by numbers, starting from the left (ch. 1, ch. 2, ch. 3, etc.). Some gestures are effectuated with one hand (unimanual chirograms) and others with both hands (bimanual chirograms). Typical bimanual gestures are, for example, hands crossed on the chest—annotated with the symbol ɣ, here in ch. 1—and the joined palms—annotated with the symbol Λ, here in ch. 3. Other characters are making two different unimanual gestures and therefore appear with two different symbols in the chiroscript, such as hand on the chest and the fist—૧○, here in ch. 7. Therefore, each figure with visible hands is represented either with one bimanual symbol or two unimanual symbols. In some cases, one of the hands may not be visible. Then, the chiroscript displays only one unimanual symbol, as in ch. 8. All the characters are separated with at least three inter-word spaces.

Every iconographic topic is characterized by a set of co-occurrent gestures typical to it, which constitute a major part of the accord. For example, here, typical co-occurrent gestures are the benedictio, the pointed forefinger, the hand on the chest, and the open palms forward. Along with the co-occurrent chirograms, painters may introduce atypical or even dissonant gestures. The signum harpocraticum—the finger on the lips, asking for silence—in Champaigne's "Last Supper", (ch. 9—ʔʝ), for example, is a unique occurrence. This chirogram, typical for other topics, such as Suzanna and the Elders, Adoration of the Child, or Isaac blessing Jacob, usually never appears in the Last Supper. By incorporating this unexpected gesture, Champaigne reminds the viewer of how important the vow of silence was for those who ordered the painting—the community of Port-Royal. At the same time, the signum harpocraticum aims to calm down or interrupt the apostles, confused by the announcement of the betrayal, and to direct their attention towards the teaching Logos (Dimova 2019). The transcription reveals what co-occurrent chirograms have been used by the painter in a given composition and what unusual and unexpected chirograms he associated with the common ones.

An important aspect of the annotation is that, in the case of ambivalent chirograms (when the hand shape may look like two different chirograms at the same time), it is necessary to introduce a double characterization. This visual ambiguity or ambivalence may be on purpose, and it is important to highlight it in the annotation rather than to try to

choose the most plausible of the two gestures. In Champaigne's "Last Supper", for instance, Christ is effectuating a variation of benedictio latina with his right hand, which looks also like a finger pointed up. In the chiroscript, the double characterization appears with a slash mark: ꞁ/Į↑ (ch. 6). The chirogram benedictio is usually effectuated by extending the thumb, the forefinger, and the middle finger. Beyond being an act of benediction, it signifies the teaching of Christ Logos or, more precisely, the transposition of sacred knowledge into the living world (Dimova 2019). It is also related to the Trinity, where the number two refers to Christ. According to Dorival (1971), the mixture between benediction and pointed forefinger could refer to the double nature of Christ—half human, half divine—because the middle finger is not completely extended and the configuration could thereby display the number one and a half instead of two. Chamapaigne also introduced the ambivalent symbol "ꞁ/Į↑" in his "Annunciation" (c. 1656, Wallace Collection). The double characterization is an appropriate solution any time that there is doubt about the shape of the hand. It also helps to make the distinction from gestures which are perfectly clear.

In the chiroscript, it is also possible to see and appreciate the repetition of the same gesture more than once in a composition. This is not something obvious for the mere observation of a painting because all the hands look different. When one gesture is represented several times, it appears differently because of various points of view, shapes, orientations, locations, and so on. The advantage of the chiroscript is that the symbol will be always the same and the repetition will thereby be more explicit. In Champaigne's "Last Supper", an interesting deictic symmetry has been obtained by placing on both sides of Christ characters pointing towards him with one hand and touching or holding the table with the other. These two figures appear in the chiroscript as follows: Į→ŋf and Į←ŋf. The symmetrical deixis connects the left and the right side of the composition, emphasizing the relationships between the characters and their common commitment to the central figure. Mirroring, redundant and duplicate structures may be easily emphasized by the annotation.

The chirographic accord of "The Last Supper" by Leonardo da Vinci (Figure 4) also contains three important gestural repetitions: two peripheral and one central. Here is the transcription, in which we count 25 chirograms (24 unimanual and 1 bimanual):

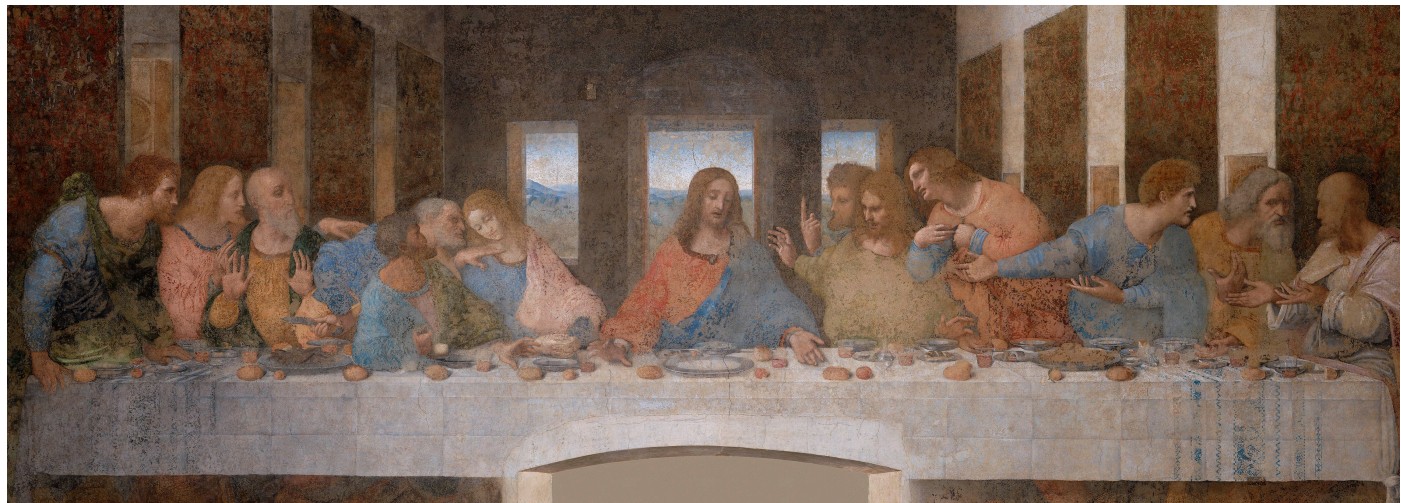

**Figure 4.** Leonardo Da Vinci, "The Last Supper", (detail), c.1495–98, tempera, 460 × 880 cm, Santa Maria delle Grazie, Milan.

ŋfŋf   ℐ€ℐ€   ẘẘ   ŋswdℐ€/Į→   ŋmoꟅgsp   §   Ʂgspꟛ   ẘẘ   Į↑ŋswd   ꝗ·ꝗ   ꟛ←ꟛ←   ꟛ/σ←ꟛ   ꟛꟛ←

The first symmetric repetition is again a deictic encircling of Christ by one character pointing at him from the left side (ch. 4) and three characters pointing at him from the right side (ch. 11, ch. 12 and ch. 13). Here again, the use of concentric deixis underlines the plural and simultaneous focus on Christ's discourse. The pointing hands of

the right side are obvious, but the one of Peter on the left side is discreet and could be missed. It is moreover integrated in another gesture—the act of putting his hand on John's shoulder—and appears in the chiroscript as a double characterization: Ⅽ/Ɪ→. The second symmetric repetition concerns the holding of knives by Peter's right hand (ch. 4) and Thomas' left hand (ch. 9). Here also, the echoing gestures only appear clearly as such in the annotation. In the actual painting, while the hand of Peter holding the knife is perfectly visible, the hand of Thomas is extremely discreet and can be noticed only by attentive observation of a high-quality digital reproduction. Holding the knife in a threatening position is an attitude used to express the vivid reaction of some apostles to the announcement of the betrayal. Positioning this gesture on both sides of Christ and combining it with a pointed forefinger (Peter points to Christ while asking John to obtain some information, and Thomas points upwards, probably invoking God) provides one more symmetrical force to the accord.

The central gestural repetition in Leonardo's "Last Supper", which is the most important one, concerns the right hand of Judas (ch. 5) and the left hand of Christ (ch. 7): they are both in a mirroring grasping position. This duplicated gesture is signified by the symbol Ş$^{gsp}$ in the annotation, where it is also possible to see their spatial proximity. They appear separated only by John's gesture of sorrow, the intertwined fingers. The symmetric grasping position of both hands refers to the main clue about the betrayal, indicated in the scriptures: "The one who has dipped his hand into the bowl with me will betray me" [Mt 26–23]. Leonardo's decision to represent Christ's right hand with this particular configuration, rather than in benedictio or in holding the bread, is not only unprecedented but stands against the iconographic tradition. This unique occurrence will never be reproduced in posterior treatments of the Last Supper episode. Importantly, to be *about* to grasp the same meal signifies the inextricable relationship between the act of betrayal and the subsequent eucharistic ritual (Polzer 2011; Dimova 2019). Nevertheless, the dramatic focus on this groundbreaking detail may remain discrete while observing the whole composition. This is an example of how a central iconographic detail can be visualized and highlighted by the transcription. The inherent rhythmicity of the chirographic accords (repetitions, duplications, echoes,) appears more visibly.

## 6. Ghirlandaio's Variations

Domenico Ghirlandaio has represented the Last Supper at least four, possibly five times (Hostetter 1991). Three of these representations have been preserved, namely in the abbey of San Michele Arcangelo in Passignano (1476), in the church Ognissanti in Florence (1480), and in the church San Marco in Florence (1486). Comparing the whole sequences of gestures attributed to the apostles and Christ in these three frescoes will give us a more precise vision of how they relate to each other.

In Table 1, the chirographic accords of the three frescoes are superposed and colored according to the dominant semantic categories. When both hands belong to different semantic categories, only the color of the more important one is shown.

**Table 1.** Comparative table of the chirographic accords in the Last Suppers by Domenico Ghirlandaio according to the dominant semantic categories.

|   | n/a | n/a | n/a | n/a | Peter | Christ | John | Judas | n/a | n/a | n/a | n/a | n/a |
|---|---|---|---|---|---|---|---|---|---|---|---|---|---|
| P | Ψ/ẘ | ɣ | ꧁$^{swd}$Ş$^{hold}$ | ꞁꝫ | ꧁$^{swd}$Ꝉ | ꞁ/Ψ | ꝙ | Ş$^{hip}$ | Ş$^{acro}$/ɣ | ꝫꝄ | Ş | Ꝉ꧁$^{swd}$ | Ψ/ẘ |
| O | ꧁$^{f}$ʔ | Ɪ→꧁$^{f}$ | ꧁$^{f}$ | ꝙꝏ | ꧁$^{swd}$σ→ | ꞁ | ɣ | ꧁$^{mo}$ | Ş | ꞁ꧁$^{f}$ | ꝙꝫ | ꧁$^{f}$ꝙ | ꝙꝙ |
| M | ɣ | ꝏ | Ɪ̰→꧁$^{f}$ | Ş | ꧁$^{swd}$꧁$^{f}$ | ꞁ | Ꝉ/ɣ | ꧁$^{br}$ | Ş$^{hold}$ | Ş | ꧁$^{br}$꧁$^{br}$ | ẘ꧁$^{f}$ | ꝙꝙ |

P—Passignano; O—Ognissanti; M—San Marco; Discursive ▉ Emotional ▉ Devotional ▉ Postural ▉ Manipulating ▉ .

It was argued that in the iconography of the Last Supper, devotional, discursive and emotional gestures have often been reunited since the 15th century in order to express the ambivalence of this biblical scene, in which the announcement of the imminent betrayal is

juxtaposed to the celebration of the Eucharist and—in some cases—to the Sermon of Christ (Dimova 2019). Usually, if the painting is intended to be placed in a profane space, such as a refectory, the accent will be put on the betrayal, whereas if it is to be displayed in a sacred space, such as an altar in a church or a chapel, the celebration of the Eucharist must be dominant (Gilbert 1972). This rule is not strictly respected, though. The examination of gestures offers a supplementary and more accurate appreciation of the painter's intentions. If most of the chirograms are part of the devotional repertoire, the Eucharist is the main focus of the painting. But if discursive and emotional gestures dominate, then the betrayal announcement is more emphasized because the apostles react to that either by asking questions and discussing or by expressing their pain, astonishment and anger.

In Table 1, the distribution of the semantic categories shows a slightly different orientation in the three versions, even though they are all destined to a refectory space. The Last Supper in Passignano contains six characters in devotional attitudes, only three with discursive, and two with emotional gestures. By contrast, the Last Supper of Ognissanti displays six characters in discursive attitudes, three devotional, and two emotional. The last version of the topic, in San Marco, is less polarized, with dominant inner emotional states. From this general glance at the chiroscript, we can make our first observation: Ghirlandaio opted for a different distribution of the general semantic categories, starting with a predominantly devotional image, then highlighting the discursive vivid aspect of the story, and finally turning towards an inner emotional dimension. Is this observation matching the existing historical commentaries of the three Last Suppers? According to Rachel Hostetter, the first and the last versions are indeed quiet and contemplative, while the Ognissanti version—the most famous one—is much more agitated, naturalistic, and dramatic. The apostles are expressing "dismay and puzzlement" both via faces and gestures, (Hostetter 1991). Defining the dominant semantic categories is indeed an important step in the gestural characterization of a particular painting, especially when the chirograms are unclear or disruptive according to the iconographic tradition. Even though every topic has a specific main theme or "mood", it is always possible for a painter to lean more towards one or the other semantic aspect of this main theme. For the 17th century, this type of analysis could cross the theory of the modes, particularly promoted by Nicolas Poussin (Testelin 1693). Perhaps, weighting the semantic categories in each composition would help to readjust some well-known notions of art theory.

In Passignano, we can see the highest rate of double characterizations, which means that the gestures are ambiguous or not well determined. On the other hand, in Ognissanti, the attitudes are much clearer and some typical for the topic discursive chirograms appear, such as the self-indicating hands on the chest and the pointing finger. The discursive gestures are also framing both sides of the linear composition in Ognissanti; in the same way as in Passignano, devotional chirograms stand on both ends of the table. Peter, Christ, John, and Judas are positioned in the exact same order in the three versions and are the only ones that can be identified with certainty because of their attributive gestures. Being the constant pillars of this topic, they set the tone of the whole accord in terms of coherence or contrasts. The devotional *benedictio* of Christ does not have the same value in Passignano and in Ognissanti. In the former, it is combined with five other devotional gestures in a more Eucharistic-oriented harmonious accord, while in the latter, it contrasts with the surrounding discursive gestures and agitation. In addition, it exhibits a temporal divergence between Christ, already involved in the Eucharistic ritual, and the disciples still trapped in the astonishment of the predicted betrayal. Chirograms do not only have a proper meaning but also a temporal dimension in the sense that they might belong to different moments of the story and still appear together in the composition. In other words, some accords may be chronologically coherent and others chronologically divergent.

To conclude this overview of Ghirlandaio's variations, we can propose a translation of the most prominent of the three accords—the one in Ognissanti (Figure 5)—by attributing a verbal meaning to each gesture, defined both by the pictorial context and the biblical passages. The words corresponding to the gestures are underlined, and around them, there is

a hypothetical reconstruction of a sentence. From left to right, here is what the characters would say according to their gestures and face expressions:

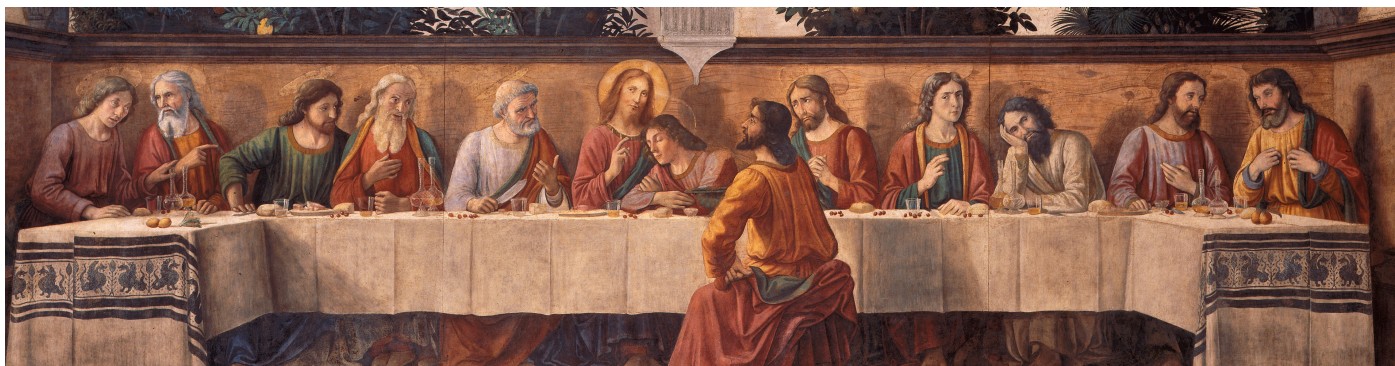

**Figure 5.** Domenico Ghirlandaio, "Last Supper", (detail), 1480, fresco, 400 × 810 cm, Ognisanti, Florence.

| | |
|---|---|
| ꟼ₅ᶠ⁆ | "I'm quiet and, by touching your wrist, I ask you to be so too." |
| I̠→ꟼ₅ᶠ | "Yes, but have you heard what He said"? |
| ꟼ₅ᶠ | "I remain passive but lean forwards to listen" |
| ꟼℴ | "The real faith is in the heart, Judas. I'm indignant" |
| ꟼ₅ˢʷᵈσ→ | "I'm angry and wonder if you will betray Him, Judas?" |
| Ⴑ | "I am the Word and I bless you, beholder". |
| ɣ | "I am fervently devoted and melancholic". |
| ꟼ₅ᵐᵒ | "I am carrying the moneybag, my reward for Christ's denunciation." |
| § | "I'm in sorrow" |
| Ⴑ/ꟼꟼ₅ᶠ | "I recreate the blessing sign of the Lord, beholder". |
| ꟼ⊟ | "I'm melancholic, beholder" |
| ꟼ₅ᶠꟼ | "I'm quiet, but maybe it's me?" |
| ꟼꟼ | "Is it me, Lord?" |

The exact translation of every posture is of course a matter of a subjective historical interpretation. However, studying and considering the meaning of each chirogram may refine our understanding of the pictorial language beyond iconographic stereotypes. The gestures do not always exactly match what is expected to be seen based on the reading of biblical sources. They provide a good foundation for revealing the highly subjective interpretation of the painters themselves as well as the specific aspects of the story that may have count for the patrons. In the Last Suepr of Ognissanti, three of the characters direct their gaze straight towards the beholder, engaging them in a profound reflection about the mystery of grace and mercy. As emphasized by Hostetter (1991), the differences between the three versions of the Last Supper by Ghirlandaio can be explained to some extent by the commissioning communities. The fresco in Ognissanti was made for the Frati Umiliati of the wool guild. They were Benedictine craftsmen, turned towards the poor and the humble, who would appreciate direct and natural attitudes of the depicted characters. On the other hand, the fresco in Passignano was commissioned by the much more intellectual order of the Dominicans. Consequently, the discursive aliveness in Ognissanti was expected to inspire transparent ideas in the beholder's mind to a much stronger degree than the devotional quietness in Passignano would. Whether this is truly fact or not may only be revealed by empirical approaches of art perception, tailored to semiotics. This is the final and essential step in the study of pictorial gestures.

## 7. Limits, Applications, and Interdisciplinarity of the Chiroscript

There are many instructive lessons we could learn from the history of transcription tools for dynamic visual languages. Since it is difficult to design a well-functioning graphical representation of these languages (because of their three-dimensional moving aspects),

the transcription systems that have been created are often criticized for their complexity and impracticality (Kato 2008). In order to be shared and benefit the scientific community, a transcription system must be non-ambiguous and relatively easy to use. This is also the case, even to a greater extent, for the writing systems. Outside of academic circles, sign languages all over the world still do not have officially and generally accepted writing systems despite of the existence of well-known projects such as the SignWriting of Valerie Sutton. Therefore, if we introduce a transcriptional procedure for the study of chirographic accords, we need to be particularly vigilant to its usability. Generally, a good level of usability is achieved only when a particular system benefits from multiple users and tests. The goal of this paper is to propose a new transcription system which could be tested and eventually improved by its users.

The stylistic and semantic variety of paintings may lead to subjective transcriptions which lack alignment with each other. Again, this problem could only be addressed by practice, and it does not in and of itself call into question the need for a transcription system. One way to reduce the risk of subjective and inexact identifications of pictorial gestures is to systematically use the aforementioned double characterizations.

Another challenge to have in mind is the linearity of the chiroscript as opposed to the complex spatial structures of the paintings. As we mentioned before, multiple planes would impose to the transcriber to follow the characters in a zigzag or circular order, and they will ultimately appear in the chiroscript as if they were in a friezelike composition. For the moment, we consider that it is preferable though to keep this constraint instead of adding a supplementary parameter in the chiroscript related to the spatial position of the characters. This may overcharge the system without much benefit. Moreover, the goal of the transcription is to create a simplified, schematic, and objective version of pictorial gestures in order to allow their structural analysis. A too-detailed transcription could dilute the focus, just as the entire painting does with its multiplicity of components and effects. Extracting the accords from the composition and presenting them as a separate entity helps to obtain an objective analytical view.

Finally, the chiroscript is not intended to work as an autonomous and self-sufficient writing, but only as a supporting, concomitant tool of analysis. Of course, it has to be combined with the directly related face expressions and postures and consequently to all the other elements of the painting.

The chiroscript can be applied to different types of studies and on different scales. On the level of multiple paintings, it can reveal important commonalities or differences between iconographic topics, painters, and periods. For example, if we want to know what the co-occurrent chirograms in the topic of Jesus and the woman taken in adultery are, we can transcribe multiple representations of this pericope and compare the transcriptions via statistical methods. It will then be possible to know what are the three or four chirograms which appear the most frequently together in this topic. Co-occurrent chirograms represent an important iconographic signature which had never been analyzed so far. They are also one of the foundations of the pictorial grammar collectively created by the painters via influences and quotations. Another type of analysis could be the comparison of all the chirographic accords created by a particular painter regardless of the subject matter. Some combinations of chirograms may be typical of that painter. On a much larger scale, it would be also interesting to compare the accords of many paintings from a particular style/period, all topics included, in order to emphasize which combinations of chirograms are the most frequent and/or featured ones. Pictorial styles and periods have gestural delineations, but these are difficult to see with traditional art historical approaches. Only large-scale statistical methods could unveil them. This is the logical next step in the development of the chiroscript: comparing the transcriptions of iconographic corpuses of interest and provide new statistical insights and deeper knowledge about the pictorial gestures.

So far, the chiroscript implies only manual transcription effectuated by an expert of pictorial chirograms. An interesting perspective would be the combination of our method with technologies for gesture recognition in art. In a recent study, a computational

method of recognition of nine canonical chirograms was established and resulted in the creation of a dataset with around 1400 correctly identified and annotated hand images (Bernasconi et al. 2023). Based on 2D hand keypoint features and HPE (human pose estimation), this technology fills an important gap in the study of chirograms and opens the way for new measures and enquiries. It was for example possible to calculate the almost equal distribution between left and right hand for the majority of the unimanual gestures except benedictio, pointing index, and hand on the chest where the right hand dominates. An interesting insight was also given into the frequency of the gestures in general. If the pointing index and the open hand forward are expected to come in the two first positions, the following chirograms, namely hand on the chest and praying hands, are much more unpredictable. Formal analyses were also effectuated to underline visual similarities and rates of confusion between the chirograms.

As stated by the authors, the recognition of pictorial hand gestures is technically very challenging. This is mainly due to the stylistic varieties and the visual ambiguities we mentioned before. For example, the benedictio chirogram can sometimes look like an open hand, a pointed forefinger, and/or a dactylological chirogram, as we saw in Champaigne's Last Supper. Champaigne created an ambivalent hand shape which refers mainly to the benedictio because of the context but also to the number two and a half and to the forefinger pointed up. This same gesture was used also in his "Annunciation" (c. 1656, Wallace Collection), where it refers mainly to the pointed forefinger and the number two and a half but not to the benedictio. Synonyms, homochirograms, and ambivalent gestures in painting can be particularly complicated for computer recognition. They would require an expert manual annotation. Nevertheless, as was demonstrated in Bernasconi's study, there are many gestures which can be correctly recognized and collected. Computer recognition could significantly improve the art historical analysis of pictorial gestures, providing the possibility of analyzing vast collections and revealing unknown typological and statistical patterns. Although it is not possible for the moment to cover the totality of the iconographic gestures by automatic methods, semi-automatic recognition and transcription would be a valid approach as well, complementary to traditional art historical practices. The chiroscript could be associated with HPE methods in order to establish more complete analytical datasets.

In addition to being closely related to digital humanities, the chiroscript could also nourish the connections between art history and cognitive sciences. As we mentioned briefly, the final step in the study of pictorial chirograms is the consideration of their perception by different kinds of beholders. During the last two decades, the implementation of eye tracking technologies and psychological inquiries in art history raised awareness about the multiplicity of factors influencing art perception. Language and culture in particular play an essential role in viewing habits and the interpretation of paintings (Klein et al. 2014; Rosenberg 2016; Brinkmann 2021; Reitstätter et al. 2022; Brinkmann et al. 2023). Studying the influence of chirograms on eye movements as well as their contribution to the understanding of pictorial narratives is therefore a logical continuation for these empirical approaches. The chiroscript offers the possibility of establishing a rigorous framework for experimental studies. Are there some chirographic accords which are better perceived or understood than others? Are the eye movements consistent with the internal organization of the chirograms? How do the beholders put an order into the simultaneous "speaking" of all characters? These and other questions could be answered by the implementation of systematic and objective analyses of the chirographic accords into empirical art historical studies.

Finally, even though the chiroscript is tailored to Early Modern painting, it could be extended and adapted to other types of images and periods since it is essentially devoted to two-dimensional gestures. Medieval illuminations, narrative book illustrations, didactic plates, and even 20th century advertising and documentary pictures could be covered by this method of transcription. Linguistic studies dedicated to historical annotations and representations of sign languages could also benefit from the transcriptional analysis of art pic-

tures since both deal with the fundamental question of how a dynamic three-dimensional hand movement could be graphically portrayed.

## 8. Conclusions

The chiroscript provides a permanent written code for each depicted chirogram, which had the advantage of being much shorter and functional than the typical verbal descriptions. Importantly, this code possesses an iconic quality. Pictorial chirograms form a one-of-a-kind linguistic modality which could be much better appreciated if it had its own graphical codification. The chiroscript will help to emphasize the epistemological and cultural uniqueness of the pictorial language.

Chirograms are of course just a stage, but an important one, especially concerning the intelligibility of art. As Bergesen says: "A study of language-like features of paintings seeks to make the unintelligible in all styles intelligible" (Bergesen 2000). Implementing a transcription system into the study of pictorial gestures would elucidate both the syntax and the meaning of the accords. Highlighting the co-occurrence, the ambivalence, the deictic structures, and the predominant semantic categories of the chirograms could better reveal their semiotic qualities and even lead to systematic translation. Therefore, the chiroscript method could contribute to making paintings more accessible and transparent.

**Supplementary Materials:** The following supporting information can be downloaded at: https://www.mdpi.com/article/10.3390/arts12040179/s1, User guide; Chirokey Keyboard Package (https://osf.io/8g4fb/).

**Funding:** This research received no external funding.

**Data Availability Statement:** No new data were created or analyzed in this study. Data sharing is not applicable to this article.

**Acknowledgments:** Open Access Funding by the Austria Science Fund (FWF).

**Conflicts of Interest:** The author declare no conflict of interest.

## Appendix A. Semantic Table of the Chirograms

| Discursive | |
|---|---|
| Gestures of argumentation and knowledge, deictics, numbers of dactylology, speaking, message delivering | |
| Finger pointing at one's eye | Little finger folded (Number 1) |
| Forefinger pointed at the forehead | Palm up—palm down |
| Forefinger on the mouth (Signum harpocraticum) | Open hand |
| Forefinger pointed in someone else's palm (Fortune gesture) | Pointed forefinger |
| Forefinger pointed at the palm of the other hand | Pointed thumb |
| Forefinger pointed up | Self-pointing hand(s) |
| Forefinger touching the middle of the thumb (Number 10 or 1000) | Thumb and forefinger joint (*Elocutio*) |
| Hand on someone's shoulder | Thumb and middle finger joint |
| Hold or touch a finger of the other hand (Comput digitis) | Thumb folded (Number 50 or 5000) |
| Little finger and ring finger folded (Number 2) | Touch the top of one's head |
| **Emotional** | |
| Gestures of moral or physical pain, astonishment, anger, strong stimulation of the senses | |
| Arm up—arm down | Head leaning on the hand (Melancholy) |
| Bite one's finger or hand | Hide an eye |
| Cover the ears with the hands | Intertwined fingers |
| Cover the nose with the hands | Palms forwards |
| Fist | Pull one's hair out |
| Gesture of glare protection | Touch or hold one's beard |
| Grasp or hold someone else's wrist | Wiping one's tears |
| Hand covering the face | |

| **Devotional** | |
|---|---|
| *Gestures of prayer, adoration, veneration, spiritual meditation* | |
| Crucified arms | Hand(s) on the chest |
| Embrace the base of the cross | Joined palms |
| Exhibit the stigma | Lifting the veil |
| Forefinger and middle finger extended (Benedictio latina variation) | Open hands raised up (*Ergebenheitsgestus*) |
| Hand on the chest, the other raised up (*Gestus der Beteuerung*) | Raising up the Host |
| Hand holding or touching a skull (Vanitas gesture) | Thumb and ring finger joint (Benedictio graeca) |
| Hands crossed on the chest (*Inbrunstgestus*) | Thumb, forefinger and middle finger extended (Benedictio latina) |
| **Ritual** | |
| *Gestures of oath, pact, allegiance, benediction* | |
| Arm forwards (Roman salute) | Hand on the Bible |
| Break a staff on one's knee | Handshake (Dextrarum junctio) |
| Gesture of washing its hands | Laying on of hands (Imposition) |
| Hand on someone else's genitals (Sub femore) | Put the wedding ring |
| Hand on someone else's thigh (Sub femore) | Uniting the hands of a marrying couple (Iunctor gesture) |
| **Insulting** | |
| *Gestures of insult, mocking, rejection, conjuration, apotropaic acts, amusement* | |
| Blow one's nose | Hand in the elbow pit (Manichetto) |
| Fist in the palm of the other hand | Hand slipping underneath the chin (Negativa) |
| Forefinger and middle finger extended forwards or downwards | Little finger in the ear |
| Forefinger and little finger extended (Horns) | Middle finger extended |
| Forefinger and ring finger extended (Horns variation) | Stretch out the mouth with the hands |
| Forefinger inserted in the other o-shaped hand | Thumb inserted between the forefinger and the middle finger of a closed hand (Fica) |
| Forefinger on the nose | Thumb on the mouth |
| Forefinger through an open mouth | Thumb on the nose |
| Forefinger, middle finger and ring finger intertwined, while the other hand pushes the little finger (Jester gesture) | Thumb pointed up |
| Forefingers crossed in the air | Twisted or deformed fingers |
| Forefingers extended next to each other (Horns variation) | Wrists crossed and raised up with outstretched fingers |
| **Postural** | |
| *Gestures of identity, status, virtues and vices, elegance, manners, attentive listening/examination* | |
| Arm of dead Christ hanging inert | Hands holding one another in the back |
| Crossed arms | Hands holding one another |
| Crossed wrists | Hold and look through glasses |
| Fingers inserted between the pages of a book (Bookmark gesture) | Hold two or three fingers of the other hand |
| Hand hidden in the jacket | Little finger extended |
| Hand hiding the genitals (Venus pudica, Eve puduca) | Middle finger and ring finger stuck together |
| Hand on the cheek or chin | Take off one's hat |
| Hand on the ear to hear | To be about to grasp something |
| Hand on the hip | To beg |
| **Manipulating** | |
| *Manipulation of symbolic and other objects* | |
| Hold a book or a sheet of paper | Hold a staff |
| Hold a bread | Hold a sword, a knife or another weapon |
| Hold a crown | Hold a vase, glass, bottle, recipient |
| Hold a fleur-de-lys | Hold an animal |
| Hold a hat | Hold coins or moneybag |
| Hold a light, a candle, a torch | Hold the Crucifix |
| Hold a martyrdom palm branch | Hold the globus cruciger |
| Hold a mirror | Playing a musical instrument |
| Hold a piece of cloth or drapery | Put one's hand on a table/furniture |
| Hold a set of cards | |

## Appendix B. Table with the 26 Symbols of the Chiroscript

| Chiroscript | | | | | | | |
|---|---|---|---|---|---|---|---|
| | Chirogram | | Symbol | Unicode | | Chirogram | Symbol | Unicode |
|  | Կ | Open hand | Armenian capital vew | 054E |  | ∞ | Dextrarum junctio | Infinity | 221E |
|  | Ị | Pointed forefinger | Latin capital I with tilde below | 1E2C |  | Ѥ | Hand on someone else's shoulder or body | Cyrillic capital iotified e | 0464 |
|  | Ֆ | Comput digitis | Armenian capital feh | 0556 |  | Գ | Hand on the chest | Armenian capital gim | 0533 |
|  | ɟ | Signum harpocraticum | Latin small dotless j with stroke | 025F |  | Ն | One hand on the chest, the other extended | Armenian capital now | 0546 |
|  | և | Benedictio | Armenian small ligature ech yiwn | 0587 |  | ɣ | Hands crossed on the chest | Latin small gamma | 0263 |
|  | σ | Pointed thumb | Greek small sigma | 03C3 |  | Ψ | Elevated hand(s) | Greek capital psi | 03A8 |
|  | ƴ | Forefinger and middle finger extended | Latin small y with hook | 01B4 |  | Λ | Joined palms | Greek capital lambda | 039B |
|  | Ձ | Ring gesture | Armenian capital ja | 0541 |  | Ϫ | Horns gesture | Coptic capital gangia | 03EA |
|  | § | Intertwined fingers | Section sign | 00A7 |  | ճ | Fig gesture | Armenian small cheh | 0573 |
|  | ẘ | Palm(s) forwards | Latin small w with ring above | 1E98 |  | ҧ | Manipulation of an object | Cyrillic small pe with middle hook | 04A7 |
|  | ᴒ | Fist | Latin small sideways open o | 1D12 |  | ᴝ | Hand in rest | Latin small sideways u | 1D1D |
|  | ϥ | Hand on the cheek or head | Coptic capital fei | 03E4 |  | Ș | Any specific gesture | Latin capital s with swash tail | 2C7E |
|  | ʡ | Grasp or hold someone else's wrist | Latin glottal stop with stroke | 02A1 |  | Ͽ | Any unknown gesture | Greek small reversed dotted lunate sigma | 03FF |

## Appendix C. Symbols for Chirograms without Assigned Character in the Chiroscript

### List of specific gestures « Ş »

| | | | |
|---|---|---|---|
| Ş$^{arm}$ | Arm forwards (Roman salute) | Ş$^{chin}$ | Hand on the cheek or chin |
| Ş$^{inert}$ | Arm of dead Christ hanging inert | Ş$^{hear}$ | Hand on the ear to hear |
| Ş$^{epi}$ | Arm up—arm down | Ş$^{hip}$ | Hand on the hip |
| Ş$^{bit}$ | Bite one's finger or hand | Ş$^{kn}$ | Hand on the thigh or knee |
| Ş$^{blow}$ | Blow one's nose | Ş$^{nega}$ | Hand slipping underneath the chin (Negativa) |
| Ş$^{break}$ | Break a staff on one's knee | Ş$^{holdb}$ | Hands holding one another in the back |
| Ş$^{ears}$ | Cover the ears with the hands | Ş$^{hold}$ | Hands holding one another |
| Ş$^{nose}$ | Cover the nose with the hands | Ş$^{eyeh}$ | Hide an eye |
| Ş$^{acro}$ | Crossed arms | Ş$^{glas}$ | Hold and look through glasses |
| Ş$^{wcro}$ | Crossed wrists | Ş$^{three}$ | Hold two or three fingers of the other hand |
| Ş$^{cross}$ | Crucified arms | Ş$^{ben}$ | Laying on of hands (Imposition) |
| Ş$^{emcro}$ | Embrace the base of the cross | Ş$^{veil}$ | Lifting the veil |
| Ş$^{stigm}$ | Exhibit the stigma | Ş$^{n2}$ | Little finger and ring finger folded (Number 2) |
| Ş$^{eye}$ | Finger pointing at one's eye | Ş$^{little}$ | Little finger extended |
| Ş$^{bm}$ | Fingers inserted between the pages of a book (Bookmark gesture) | Ş$^{n1}$ | Little finger folded (Number 1) |
| Ş$^{rab}$ | Fist in the palm of the other hand (*Rabia*) | Ş$^{auri}$ | Little finger in the ear |
| Ş$^{v}$ | Forefinger and middle finger extended forwards or downwards | Ş$^{101}$ | Middle finger and ring finger stuck together |
| Ş$^{hor}$ | Forefinger and ring finger extended (Horns variation) | Ş$^{mid}$ | Middle finger extended |
| Ş$^{fo}$ | Forefinger inserted in the other o-shaped hand | Ş$^{updo}$ | Palm up—palm down |
| Ş$^{fno}$ | Forefinger on the nose | Ş$^{hair}$ | Pull one's hair out |
| Ş$^{forh}$ | Forefinger pointed at the forehead | Ş$^{ring}$ | Put the wedding ring |
| Ş$^{fplm}$ | Forefinger pointed at the palm of the other hand | Ş$^{host}$ | Raising up the Host |
| Ş$^{fortun}$ | Forefinger pointed in someone else's palm (Fortune gesture) | Ş$^{mo}$ | Stretch out the mouth with the hands |
| Ş$^{fingo}$ | Forefinger through an open mouth | Ş$^{hat}$ | Take off one's hat |
| Ş$^{n10}$ | Forefinger touching the middle of the thumb (Number 10 or 1000) | Ş$^{ring}$ | Thumb and middle finger joint |
| Ş$^{jester}$ | Forefinger, middle finger and ring finger intertwined, while the other hand pushes the little finger (Jester gesture) | Ş$^{begr}$ | Thumb and ring finger joint (Benedictio graeca) |
| Ş$^{x}$ | Forefingers crossed in the air | Ş$^{n50}$ | Thumb folded (Number 50 or 5000) |
| Ş$^{hoff}$ | Forefingers extended next to each other (Horns variation) | Ş$^{thumb}$ | Thumb on the mouth |
| Ş$^{glr}$ | Gesture of glare protection | Ş$^{pied}$ | Thumb on the nose |
| Ş$^{wash}$ | Gesture of washing its hands | Ş$^{gsp}$ | To be about to grasp something |
| Ş$^{covef}$ | Hand covering the face | Ş$^{beg}$ | To beg |
| Ş$^{jack}$ | Hand hidden in the jacket | Ş$^{bd}$ | Touch or hold one's beard |
| Ş$^{pud}$ | Hand hiding the genitals (Venus pudica, Eve puduca) | Ş$^{toph}$ | Touch the top of one's head |
| Ş$^{skull}$ | Hand holding or touching a skull (Vanitas gesture) | Ş$^{twist}$ | Twisted or deformed fingers |
| Ş$^{man}$ | Hand in the elbow pit (*Manichetto*) | Ş$^{u}$ | Uniting the hands of a marrying couple (Iunctor gesture) |
| Ş$^{sufg}$ | Hand on someone else's genitals (Sub femore) | Ş$^{tear}$ | Wiping one's tears |
| Ş$^{suft}$ | Hand on someone else's thigh (Sub femore) | Ş$^{wup}$ | Wrists crossed and raised up with outstretched fingers |
| Ş$^{b}$ | Hand on the Bible | | |

## List of manipulation gestures « ꭓ »

| | | | |
|---|---|---|---|
| ꭓ$^{ch}$ | Hold *Arma Christi* | ꭓ$^{card}$ | Hold a set of cards |
| ꭓ$^{bk}$ | Hold a book or a sheet of paper | ꭓ$^{sff}$ | Hold a staff |
| ꭓ$^{br}$ | Hold a bread | ꭓ$^{swd}$ | Hold a sword, a knife or another weapon |
| ꭓ$^{crown}$ | Hold a crown | ꭓ$^{d}$ | Hold a tool, device, machine |
| ꭓ$^{lys}$ | Hold a fleur-de-lys | ꭓ$^{v}$ | Hold a vase, glass, bottle, recipient |
| ꭓ$^{hat}$ | Hold a hat | ꭓ$^{a}$ | Hold an animal |
| ꭓ$^{k}$ | Hold a key | ꭓ$^{mo}$ | Hold coins or moneybag |
| ꭓ$^{light}$ | Hold a light, a candle, a torch | ꭓ$^{cross}$ | Hold the Crucifix, a cross |
| ꭓ$^{palm}$ | Hold a martyrdom palm branch | ꭓ$^{glob}$ | Hold a glob or the globus cruciger |
| ꭓ$^{mir}$ | Hold a mirror | ꭓ$^{play}$ | Playing a musical instrument |
| ꭓ$^{dr}$ | Hold a piece of cloth or drapery | ꭓ$^{f}$ | Put one's hand on a table/furniture |
| ꭓ$^{q}$ | Hold a quill | | |

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

Testelin, Henry. 1693. *Sentimens des plus habiles peintres du tems, sur la pratique de la peinture et sculpture [...]*. La Haye: Matthieu Rogguet, s.d.

