# Peer review of "Chiroscript: Transcription System for Studying Hand Gestures in Early Modern Painting"

_arts, 2023_

Round 1
Reviewer 1 Report
Please read the attachment. Thank you.

Reviewer 2 Report
The paper represents an interesting contribution to the systematic study of pictorial gestures.
Some more specific comments for potential improvement of the paper:
11) The reader would better understand the chirographic accords of the three frescoes if the figures of all the three frescos would also be included in the paper
22) In general, the paper would benefit from including more examples of artwork images and the corresponding transcription. In particular, to also showcase and elaborate on the potential limitations of this system, as well as of the question of applicability within different contexts.
33) Also, because the image serves a reference to understand the transcription, it would be good to move Fig 1 before the hypothetical sentences.
44) Would be good to elaborate a bit more on the analytical potential of chiroscript as a tool, what is the scope of potential applications, what are the envisioned next steps? In particular in relation to the scale of analysis (e.g. how this type of transcription can foster analysis of pictorial gestures on a large collection of artworks)
55) In relation to the previous comment, it seems to me worth mentioning that just recently a paper has been published which analyses the role of gestures in the context of digital humanities (Bernasconi, V.; Cetinić, E.; Impett, L. A Computational Approach to Hand Pose Recognition in Early Modern Paintings. J. Imaging 2023, 9, 120), perhaps it might be worth referencing it and commenting on potential future integration of chiroscript with digital tools
6
The style of writing could be improved (avoid too many comma splices)
Reviewer 3 Report
The paper seems to be a description of some aspects regarding chiroscript and visualisation of the properties of chirographic. It is not not clearly what is the proposed method.
The paper must describe clearly:
-what is the proposed method
-what is the differences between other existing ones
-what are the results
-also how obtained results can be compared with other existing results.
Round 2
Reviewer 2 Report
The authors have done a good job with the revision of the first version. I think the paper can now be accepted in its current form.
Reviewer 3 Report
Some of my comments were addressed, but some of them need improvement:
-how gestures are identified from pictures? (automatically or manually)
-what will be the benefit of the proposed method?
Round 3
Reviewer 3 Report
Since all my comments were addressed, I recommend to publish the paper.